# Long-Term Results of the Modular Physiological Wrist Prosthesis (MPW^®^) in Patients with Inflammatory Diseases

**DOI:** 10.3390/life11040355

**Published:** 2021-04-18

**Authors:** Christoph Biehl, Martin Stoll, Martin Heinrich, Lotta Biehl, Jochen Jung, Christian Heiss, Gabor Szalay

**Affiliations:** 1Department of Trauma, Hand and Reconstructive Surgery, University Hospital Giessen, 35392 Giessen, Germany; martin.heinrich@chiru.med.uni-giessen.de (M.H.); christian.heiss@chiru.med.uni-giessen.de (C.H.); szalay@praxis-sportklinik.de (G.S.); 2Orthopedic Department, Diakonie Hospital, 55543 Bad Kreuznach, Germany; pfalzklops@web.de (M.S.); Dr.med.jung@gmx.de (J.J.); 3Medical Faculty Heidelberg, Heidelberg University, 69117 Heidelberg, Germany; lotta.biehl@gmx.de

**Keywords:** wrist prosthesis, MPW^®^-prosthesis, rheumatoid arthritis, wrist function, DASH, SF-36, national public health survey

## Abstract

The wrist is among the predilection sites of over 90% of cases of rheumatoid arthritis (RA). In advanced cases, total wrist arthroplasty (TWA) is an alternative to arthrodesis. The aim of this study is to present the long-term results of the modular physiological wrist prosthesis (MPW^®^) and to match them in context with the results of a standard population survey. In a retrospective study with follow-up, patients with an MPW^®^ endoprosthesis were evaluated concerning the clinical and radiological outcome, complications were reviewed (incidence and type), and conversion to wrist fusion was assessed. Patient function measurements included the Mayo wrist score, the patient-specific wrist test, and therefore the DASH score (arm, shoulder, and hand). Thirty-four MPW^®^ wrist prostheses were implanted in 32 patients, including thirty primary implantations and four changes of the type of the endoprosthesis. Sixteen patients (18 prostheses) underwent clinical and radiological follow-up. The average follow-up time was 8.5 years (1 to 16). Poor results of the MPW prosthesis are caused by the issues of balancing with luxation and increased PE wear. Salvage procedures included revision of the TWA or fusion. In successful cases, the flexion and extension movement averaged 40 degrees. The grip force was around 2.5 kg. The common DASH score was 79 points, with limited and problematic joints of the upper extremity. The MPW wrist prosthesis offered good pain relief and functional movement in over 80% of cases. The issues of dislocation and increased PE wear prevent better long-term results, as do the joints affected. A follow-up study with fittings under a contemporary anti-rheumatic therapy with biologicals suggests increasing score results. Type of study/level of evidence: Case series, IV.

## 1. Introduction

More than 100 different diseases are counted among the diseases of the immune system, which are accompanied by inflammations of different body tissues and are mainly manifested in the articular and periarticular systems. In western nations, around 2–3% of the population suffers from these diseases. Rheumatoid arthritis (RA) is the largest group, accounting for about 0.8% of the population. If left untreated, RA not only threatens to increase mortality within the first few years compared to most types of cancer but the chronic, inflammatory, and destructive course of the disease, combined with pain, also severely restricts patients in their daily lives. RA often manifests itself primarily on the hands and wrists. Destruction of the wrists has been described in the progress of the disease in over 90% of patients. If surgical treatment is necessary, the preservation of residual mobility must be included in the planning, since arthrodesis of the wrist significantly limits its function [1]. Preoperatively, the situation of the bone and the re-constructability of the carpal height, soft-tissue balance, kinematics, and joint pivot point must be carefully examined in advance [2,3]. Recent developments in wrist endoprosthesis offer movement-preserving treatment and at the same time improve the rate of complications [4]. For rheumatics, the precarious bone situation with early onset of osteoporosis and the associated loss of bone mass has to been taken into account.

Developed as an alternative to arthrodesis, total wrist arthroplasty (TWA) not only provides pain relief but also preserves wrist movement and function. The carpal anchorage and the correct balancing of the tendons continue to be problematic. These have been the subject of continuous improvement of implants to date. The Modular Physiological Wrist prosthesis (MPW^®^) is a modularly designed, cementless, implantable Titanobium endoprosthesis (Figure 1). The special feature is the encapsulated sliding pairing of the distal olive, which is intended to imitate the mobility of the intercarpal joint line [5]. For bad bone quality, various components are available, including a coupled implant. Thus, in the case of revision surgery, it is not necessary to change all components.

The study aimed to evaluate the long-term results after endoprosthetic treatment of the wrist in rheumatics and when revision interventions became necessary. The patients were examined clinically and radiologically regarding subjective satisfaction, pain, mobility, strength in hand and wrist, and participation in activities of daily life.

## 2. Materials and Methods

Patients gave their consent in writing before inclusion in the study. A positive ethics opinion has been given (Regional Ethical Review Board for Rhineland-Palatinate; Mainz, Germany; No. 837.157.14 (9396-F)).

The retrospective study presents the long-term results of the MPW^®^ prosthesis acc. to Thabe (Link Company™, Hamburg, Germany). All patients who received the MPW^®^ wrist prosthesis between 1998 and 2005 were included. Two experienced orthopedic surgeons performed all operations. Exclusion criteria were prosthesis implantation with a wrist prosthesis other than the MPW^®^, prosthesis operation outside the time period, and post-traumatic changes (SLAC-/SNAC-wrist).

The authors included 32 patients with 34 endoprosthesis-supplied wrists in the study. Twenty-eight patients were female (29 prostheses/28 patients) and four patients were male (5 prostheses/4 patients). Patients were 39 years old on average (16 to 62 years) at the time of diagnosis of RA. The mean age at the time of surgery was 56 years (34–81 years, median 57.9 years). The mean duration of the disease was 18 years. The period between diagnosis and primary implantation was 16.4 years (2–40 years). Two patients opted for primary surgery and implantation of the MPW^®^ wrist prosthesis several years after involvement.

Twenty-six patients suffered from seropositive rheumatoid arthritis, one patient had psoriatic arthritis, and one patient had familial polyarticular chondrocalcinosis with early manifestation. Five patients were seronegative in the case of clinically clear rheumatic diathesis. One patient showed pseudarthrosis after scaphoid fracture (post-traumatic).

Thirty MPW^®^ prostheses were implanted primarily, and 4 were revision surgeries. In all operations, a previously implanted endoprosthesis (APW^®^ (anatomically physiological wrist prosthesis)) was explanted. The surgeons implanted a prosthesis 18 times on the right side and 16 times on the left side.

Twenty-five procedures were initial fittings on the wrist, eight wrists had been operated on beforehand, and one wrist had been pre-operated on several times (Table 1).

### 2.1. Basic Medication

At the time of surgery, 3 patients received cortisone alone, 7 patients regularly took Methotrexate (MTX) as monotherapy and 6 patients took MTX in combination with cortisone. Eight patients underwent a Disease-Modifying Anti-Rheumatic Drug (DMARD) therapy with other drugs (Gold, Azathioprine, Leflunomide, Chloroquine). Ten patients did not receive basic therapy at the time of implantation. None of the patients had basic therapy with Biologica at the time of surgery. Biologica were not approved for RA therapy until 2000.

Clinical and radiological follow-up examinations were performed regularly at six weeks, six months and then annually. For this study, patients underwent an additional clinical and radiological examination at the time of follow-up. Data collection was performed using different scores (Visual Analog Scale (VAS), Disabilities Of The Arm, Shoulder And Hand—Score (DASH-Score) [6] and Short-Form Health 36 (SF-36). The VAS (values 0–10) was used to measure the patient’s subjective perception of pain and to objectify it. The DASH-Score (100-point score) is primarily used to measure the function of the upper extremity. In this study, the German version 2.0 was used, without the additional modules [7]. The question about sexual activity was not asked. The SF-36 (150-point score) is a measuring instrument of health-related quality of life. The score primarily reflects the health-related quality of life (HRQoL) of the individual.

The radiological controls considered the rheumatological status classification according to Larsen, Dale and Eek [8]. In addition, the carpal height index acc. to Youm was determined, and integration of the prosthesis or loosening seams was documented [9].

### 2.2. Statistics

The study design was a case series with a level of evidence of grade IV. The primary endpoint was the survival rate of the wrist prosthesis. Secondary endpoint was the required first revision due to complications. The statistical evaluation was based on the small group, the lack of norm distribution and the non-parametric procedures with logistic regression models and the Kaplan–Meyer estimator. ANOVA and Bonferroni correction were used as control tests. Data were analyzed using SPSS v. 16.0 (IBM, Armonk, NY, USA) and with R v. 3.4.2. Collected datasets from clinical findings were explored for normality with descriptive statistics. The critical value for significance was set at *p* < 0.05. Data were exhibited in graphs as means—standard error of the mean (SEM).

## 3. Results

The main outcome parameters were the survival rate of the MPW^®^ prosthesis in situ; the number, temporal occurrence, and type of complications (infection, prosthesis loosening, dislocation, etc.); the type of surgical revision; and the subjective patient satisfaction regarding the function of the prosthesis (pain, mobility, strength), as well as the suitability for daily use. Furthermore, baseline rheumatologic therapy before surgery, a potential limitation of the other joint of the affected limb and the overall situation of the patients were evaluated.

Of the 32 patients (34 prostheses), all but one other patient could be followed up at least once. Sixteen patients (18 prostheses) underwent direct clinical and radiological follow-up during the study in the department. Eight patients had died at the time of the study, two of them within the first year. None of the deceased had complications related to the arthroplasty or as a direct result of the underlying rheumatic disease up to the time of death. Only the available data (inpatient and outpatient records as well as archived radiographs) were included in the study. In the absence of feedback, the available data were cross-checked with patients’ relatives and primary care physicians. Eight patients were unknown deceased or refused to consent to the study. The last data of these patients collected in the clinic were included in the study. This corresponds to a face-to-face responder rate of 50% and a documentation rate of 97% of surgically treated wrists (33/34). Follow-up was an average of 8.5 years (1 to 15.7 years) postoperatively.

### 3.1. Survival Rate

The survival rates of the prostheses in situ were evaluated using the Kaplan–Meier estimator and presented graphically. The time between primary implantation and first revision surgery (Polyethylene exchange, explantation, arthrodesis) was determined as the functional period of the prostheses. The median was 171 months (Figure 2). The lower limit of the confidence interval was 110 months—the upper limit could not be calculated because the number of cases was too small. At the time of follow-up, 11 of the 34 prostheses had been explanted as far as was known (33%; 11/34). The average life of the prostheses was 6.9 (1–13) years. The ‘worst-case’ survival, including the wrists lost to follow-up as a failure, was 56%.

### 3.2. Complications

During the follow-up period, 23 complications were documented (Table 2). Indirect complications must be distinguished from direct complications. Indirect complications included, for example, carpal tunnel syndrome or tendon rupture, since they were not directly attributable to the endoprosthesis. Direct complications were related to the prosthesis itself or the damage caused by it (loosening, metallosis). Early associated symptoms were mainly related to nervous problems, such as carpal tunnel syndrome. The most frequent complication (66%; 15/23) represented the failure of the polyethylene inlay, with two peaks of frequency. A first peak was seen 1.5 to 3 years after implantation, and a second peak with concomitant metal wear was seen after an average of 5.1 years. The modular design of the prosthesis allowed only the damaged PE to be replaced. Explantation was not necessary, with a stable and functioning prosthesis. One superficial and two deep infections with joint involvement occurred during the course (3/6%; 1+2/34). The superficial infection occurred together with metallosis four years after implantation. All infections were healed by two-stage revision surgery and fusion. Not a single periprosthetic fracture occurred.

### 3.3. Type of Revision

Early nerve compression syndrome (CTS, div.) was treated by neurolysis of the affected nerves. Tendon rupture could be treated by secondary suture and coupling to another extensor tendon. In cases of pure inlay damage, the PE inlays were changed, and the fixed prosthesis was left in place. In the case of inlay damage with concomitant metal wear, the vast majority of fittings required removal of the prosthesis and arthrodesis of the wrist with an angle-stable plate and grafting of a cortico-spongiosis span from the iliac crest. Infected prostheses had to be removed, and a two-stage procedure with arthrodesis was performed after sanitation of the infection.

### 3.4. Patient Satisfaction

Patient satisfaction depended on several factors, which can be represented in the analysis of the collected data. Overall, about 63% of patients rated the endoprosthetic treatment of the wrist as positive at the time as the last follow-up and would have a prosthesis implanted again.

The subjectively perceived pain and swelling decreased significantly postoperatively compared with the preoperative data. A decisive point here is the denervation of the dorsal interosseous nerve at the wrist. Definite swelling was neither reported by the patients nor seen by the investigators postoperatively.

### 3.5. Strength

In the subjective assessment of strength, 9/16 patients described a significant improvement compared to the preoperative strength (Figure 3). In the clinical assessment, extension force (in kg) and force at fist closure (in kp) were measured at the follow-up. This allowed the subjective assessment of the patients of current strength to be verified and objectified. The mean dorsiflexion force in the wrist was 2.55 kg with a median of 1.25 (standard deviation 3.86). The gross force during fist closure was measured with a balloon vigor meter and averaged 0.135 kp (standard deviation 3.47).

### 3.6. Mobility

In the clinical examination at follow-up, moderate residual wrist mobility was obtained (Table 3). Patients were grateful for the preserved movement of the wrist. Moreover, functional impairments of the neighboring joints were examined and evaluated descriptively. The most severe functional impairments were reported in the proximal joints of the fingers.

### 3.7. Scores

The data collected at follow-up were evaluated in the different score systems (DASH, SF36, VAS).

The VAS is used to measure the subjective pain sensation of the patient and its objectification. It was possible to evaluate 19 datasets pre- and postoperatively. The VAS improved from 7.0 points preoperatively to 1.8 points postoperatively.

The average DASH score was 47.1 points postoperatively, which corresponds to a poor result. At the same time, there is a pronounced spread of the measured values from 1.72 to 88.8 points.

The SF-36 does not only consider the pure prosthesis quality and function but primarily reflects the health-related quality of life (HRQoL) of the individual. A total of 16 questionnaires could be evaluated (16/34; 47%). The test takes into account concomitant diseases and additional possible limitations, not only the pure prosthesis quality and function.

### 3.8. Radiology

Radiographs of 22 wrists preoperatively and at follow-up could be compared (22/34; 64.7%) (Figure 4). Preoperative X-rays of 12 wrists were not available at follow-up. The average time between the preoperative X-ray and the last X-ray was nearly 8.5 years. Eight prostheses (23.5%) showed no evidence of loosening or other radiolucency, osteoporosis, or cyst formation with proper implant position and anatomically correct joint position. In the remaining endoprostheses, various radiological findings were found in one prosthesis. In 6 prostheses, radiological dislocation or subluxation was found in the follow-up controls (17.4%). In each case, the position of the prosthesis corresponded to that of the preoperative rheumatoid wrist. In 2 prostheses, radiolucency >1 mm was found, which corresponded to a loosened prosthesis (5.8%). In a further 9 prostheses, periprosthetic radiolucency without clinical symptoms was detected (26.5%). Natively radiologically visible osteoporosis was detected in four prostheses (11.8%). Periprosthetic cysts were detectable in 6 prostheses (17.6%).

The calculation of the carpal height index according to Youm was only calculated for the wrists with an inserted prosthesis without luxation malposition (88.9% with the inserted prosthesis or 47.1% related to the total collective) [9]. With a target value of 0.54 ± 0.03 according to Youm, the average carpal height index was 0.50 (0.27–0.61).

## 4. Discussion

The aim of this study was to evaluate and characterize the long-term outcome of the MPW^®^ prosthesis and required revision procedures. Primarily, patients were included who had an MPW^®^ prosthesis implanted at least 10 years before the start of the study in 2015. Studies on long-term results are less frequent, although the publications have increased [10,11,12]. For rheumatics, the precarious bone situation with early onset of osteoporosis and the associated loss of bone mass and accompanying pathologies of the periarticular tissues limits the conditions for the long-term success of a wrist prosthesis [13,14,15]. Considering the worst-case scenario, the survival rate of the prosthesis after 6.9 years was 44%. Compared to results of in situ wrist prostheses of other authors, namely 71–74%, this is a poor result [16,17]. Adverse events, like nerve irritation, tendonitis and tendon rupture in rheumatic patients do not necessarily have to be associated with surgery [18]. Dislocations and PE damage to the prosthesis are a major risk, associated with extensive synovitis with metallosis, leading to explantation of the prosthesis and fusion of the wrist joint [19,20].

Badge et al. and Gil et al. estimated the probability of not experiencing a revision for 8 and 15 years to be about 91% and 78%, respectively, but it was only about 30% in our cohort [21,22]. In their analysis of the Norwegian Endoprosthesis Register, Krughaug et al. were able to show a 5-year survival probability of 57% for the Elon prosthesis, while the 10-year survival for the BIAX™ was 71% [17].

In their meta-analysis, Berber et al. evaluated 24 studies with a total of 1371 endoprosthetic wrists [23]. More than 72% of the included patients had an inflammatory underlying disease, which is comparable to our population with 73.5% of rheumatic patients.

Upon comparing the different types of 4th generation prosthesis, the unique feature of the PE in the MPW^®^ prosthesis stands out. This design improves the guidance of the prosthesis, but at the same time leads to early decentralized contact of the components with increased abrasion and the risk of pathological contact and the resulting dislocation (Figure 1).

Deep infections were described in only 0.1–1.4% of cases according to meta-analysis. In our study, the proportion of deep infections was 6%. All patients had an established RA and therapy with DMARDs.

Overall, the rate of complications in the presented study was 67%, which is a poor outcome.

Concerning the results of Berber et al., the rate seems comparable. Radmer et al. reported in their study an exorbitant revision rate after an average of 4.5 years of 97.5% due to implant failure in the previous model (39/40) [24].

A decisive role for the acceptance of a surgical procedure, such as TWA, is the subjective satisfaction of the patients. It has to compete with established procedures, such as arthrodesis [1]. In addition to the pain situation, the preservation of mobility, particularly of the fingers, and improvement of strength, as well as participation in daily life, are of central importance, especially since the demands of rheumatic patients are increasing due to an improvement of new medical therapies [2]. The results of our cohort reflect a high level of patient satisfaction.

The wrist pain improved on the VAS from 7.0 preoperatively to 1.8 points postoperatively. Cooney et al. reported comparable scores from preoperative 7 to post-operative 2.3, as did Nydick et al. from 8.0 to 2.2; slightly worse scores were reported by Badge et al., with a reduction from 8.1 to 5.4 [21,25,26,27].

In terms of the range of motion, the results of our study were also comparable to those of other studies, although the range of motion in extension-flexion was limited due to the constrained guidance of the PE. Thus, prostheses with a flatter radial glide surface exhibit a greater amplitude of motion [26].

The hand strength measured during follow-up for dorsiflexion (in kg) and fist closure (in kp) correlated with the subjective assessment and satisfaction of the patients. Compared with data published in the literature, these appear poor but are of limited comparability due to differences in measurement methods [16,28].

Most wrist arthroplasty studies use the DASH score. However, this is not specific to the assessment of wrist function. Compared with other publications, the average of 79 points obtained in our patient cohort appears poor [21,25,26]. In addition, there was a widespread of scores from 1.7 to 89 points. One explanation for the significantly worse scores in the DASH score could be the advanced impairment of the wrist joint itself and the adjacent joints. In the study clinic, patients underwent endoprosthetic treatment only at an advanced Larsen stage; earlier stages were reserved for joint-preserving surgery [29]. Pfanner et al. describe an average LDE stage of 2–3 for arthroplasty [16]. The Larsen stage in our collective was 4.5. The evaluation of the radiological findings documents the described problem of (sub)dislocation and decentering of the endoprosthesis. In comparison with the overall results of Berber et al., the values are higher (osteolysis vs. prosthesis = 32.3% vs. 20.8%) and are attributed by the authors to the PE problem [23]. The vulnerability of the integration of the prosthesis is primarily due to the loosening of the carpal component, as also reported by Herzberg and Boeckstyns, among others [30].

The SF-36 is also not specific for wrist function but can be used to classify the surgery and its benefit in the overall situation of the patient. Therefore, it was collected and evaluated in addition to the VAS and DASH score. The postoperative score of the study patients is below that of the norm samples of the German general population from 1999 and 2013 (Figure 5) [31,32]. Only the scores for psychological well-being are almost identical. We attribute this to the significant pain reduction in the wrist and the improved use of the extremity in everyday life, as documented in the VAS-score.

### Limitations

The authors are aware that the study has some limitations. The number of wrists included is too small to generate robust statistical data. Preoperative score data are often incomplete. The small number of patients is also due to the fact that approximately 1/4 of the patients died before the target follow-up of at least 10 years. This leads to a worst-case scenario of 56% of lost wrist prostheses at follow-up. Statements about grip strength are also of limited value because preoperative comparisons are lacking, as are comparisons with the opposite side. The time interval between the inclusion of the hand in the disease and endoprosthetic treatment is very long in terms of international comparison, based on the restrained indication of the head of the department. However, only a relatively rough estimate of the standing time can be given.

## 5. Conclusions

The MPW^®^ prosthesis as a modularly designed fourth-generation wrist prosthesis with its encapsulated bearing couple has implant-technical peculiarities compared to other wrist prostheses. The study aimed to evaluate and present the results in the long-term course. In comparison with other publications, the results show a partially heterogeneous picture. While the subjective satisfaction of the patients and the measurable strength are comparable with the results of other studies and prostheses, the prosthesis shows clear weaknesses concerning luxation and inlay damage compared to other prosthesis models. This leads to an increased number of revisions and an increased number of secondary arthrodesis in the long-term course. Especially with respect to the better results of other wrist prostheses (e.g., Re-Motion^®^, BIAX™), the prosthesis shows deficits that stand in the way of increased use.

## Figures and Tables

**Figure 1 life-11-00355-f001:**
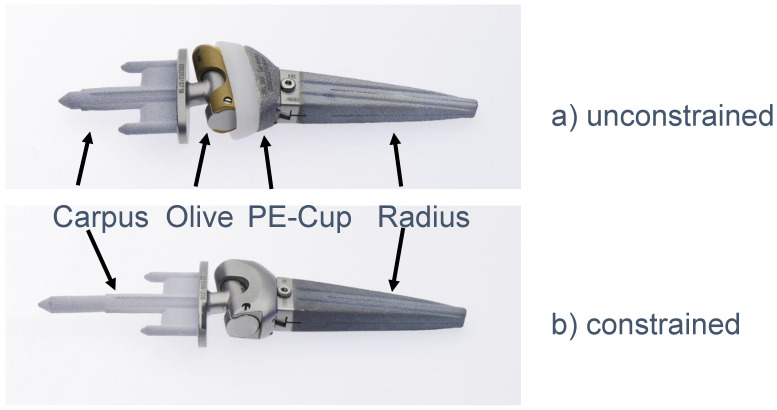
MPW^®^ Wrist Prosthesis (Link Company™, Hamburg, Germany), (**a**) unconstrained and (**b**) constrained model.

**Figure 2 life-11-00355-f002:**
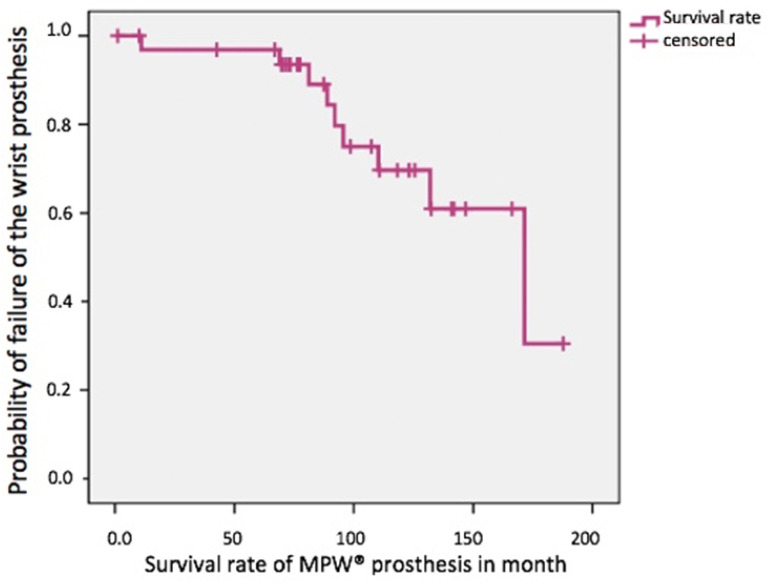
Survival rate of MPW^®^ wrist prosthesis in situ. X-axis: functional time, Y-axis: probability of failure in month (*n* = 34, events: 9, median: 171-month, 0.95 Lower CI: 110-month, 0.95 Upper CI: not applicable).

**Figure 3 life-11-00355-f003:**
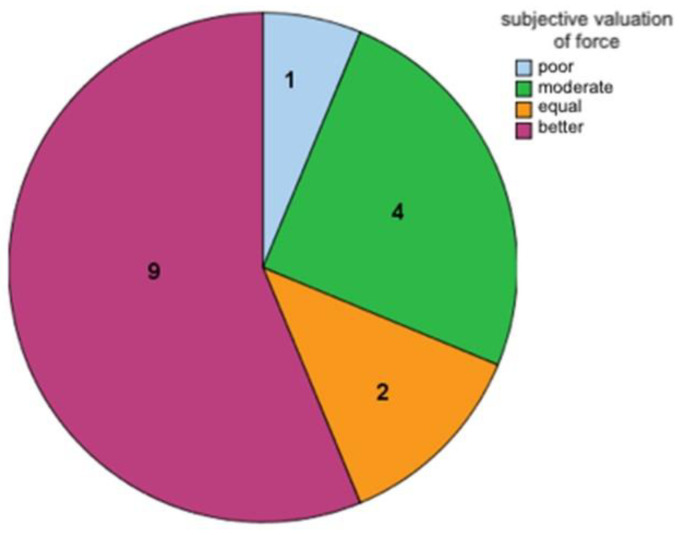
Subjective valuation of strength and force by patients (*n* = 16).

**Figure 4 life-11-00355-f004:**
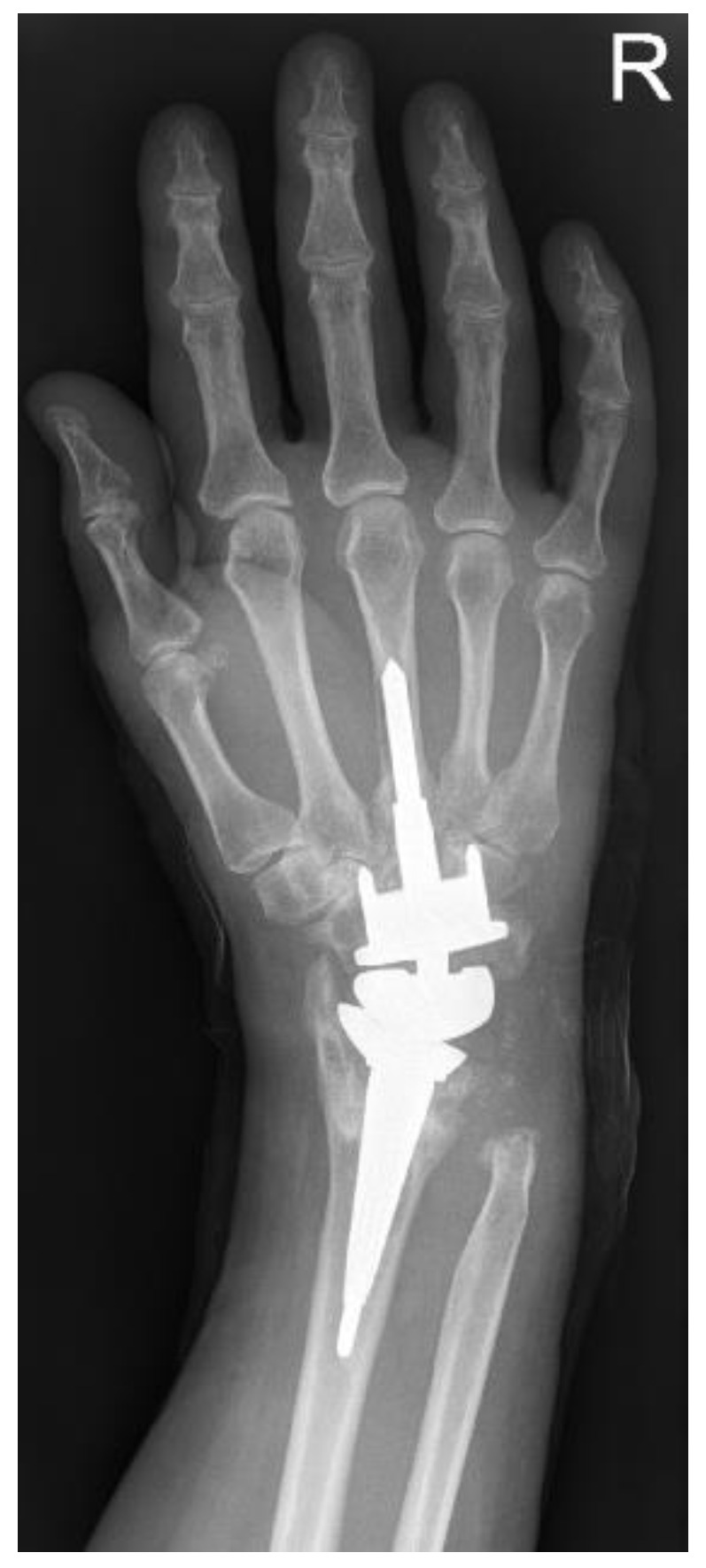
MPW^®^-prosthesis. X-ray at time of follow-up, 10 years after primary operation. The carpal component was changed after loosening (see complications).

**Figure 5 life-11-00355-f005:**
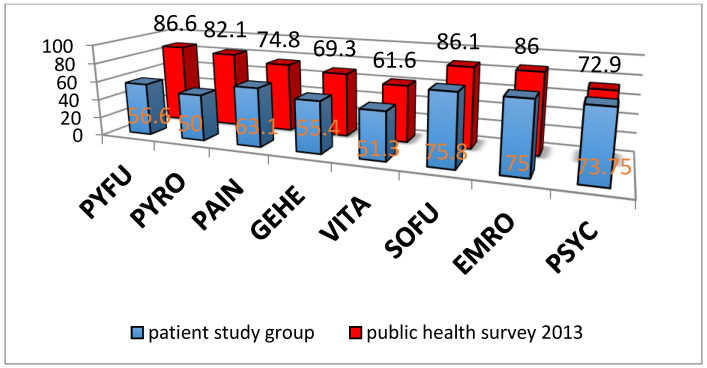
Results of the SF-36 Questionnaire. Patient study group (blue columns) vs. National public health survey (red columns) (Ellert and Kurth 2013). PYFU: physical functioning; PYRO: Role behavior due to physical dysfunction; PAIN: pain; GEHE: general health, VITA: Vitality and physical energy, SOFU: social functioning; EMRO: Emotional role function, PSYC: psychological (mental) wellbeing.

**Table 1 life-11-00355-t001:** Wrist procedures before prosthesis implantation.

	Frequency	Relative (%)
Non-Operation	25	73.5
Wrist prosthesis implantation	3	8.8
ATS / Synovectomy	1	2.9
CTS	1	2.9
Tendon transfer/reconstruction	1	2.9
Partial arthrodesis after previous operation before (ATS/Syn.)	1	2.9
Radiosynoviorthosis	1	2.9
Various previous operations: Prosthesis + ATS/Syn.	1	2.9
Total	34	100.0

**Table 2 life-11-00355-t002:** Complications/Reasons for revision in years (acc./non acc. to the MPW^®^-Prosthesis).

	N (x/34)	Relativ (%)	Mean (years)
Periprosthetic Fracture	0		
Thumb instability	1	2.9	0.87
CTS	1	2.9	0.97
(Luxation/CTS/Inlay defect)	1	2.9	1.35
Loosening carpal component	2	5.9	1.65
Deep Infection	2	5.9	1.69
Inlay defect	2	5.9	2.47
Luxation	5	14.7	2.98
(Metallosis and superficial infection)	1	2.9	4.10
Tendon rupture	1	2.9	4.17
PE-Synovitis/Metallosis	7	20.6	5.11

**Table 3 life-11-00355-t003:** Range of motion in degree, pre- and postoperative mean.

R O M (in °)	Flexion	Extension	Radial dev.	Ulnar dev.	Pronation	Suppination
preoperative	26.8	20.8	12	16.9	60	65
postoperative	26.5	12.3	25.3	9.2	58.4	79

## Data Availability

The datasets used and/or analyzed during the current study are available from the corresponding author on reasonable request.

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
