# Peer review of "Long-Term Results of the Modular Physiological Wrist Prosthesis (MPW®) in Patients with Inflammatory Diseases"

_life, 2021, doi:10.3390/life11040355_

Round 1
Reviewer 1 Report
English version:
Dear authors,
Allow me to make the following comments on the manuscript:
In my opinion, a general problem with the article is that the various parts, i.e. introductions, materials and methods, results and discussion, are often mixed up. I recommend a thorough overhaul of the work. For example, from my point of view on page 1, line 37, the remark on the increased functional claim of rheumatoid patients is part of the discussion, as justification for the indication of endoprosthesis and against arthrodisia, and should not be mentioned in the introduction. Page 2, row 54, belongs to the material and methods, as well as row 71 following.
On material and methods: I hold Figure 2 and Tab. I do not think this is necessary.
At Tab's. 2 In my view, the indication of the relative frequency is entirely sufficient.
The dash score was estimated in the work (line 122)? I do not think this approach makes sense, firstly, it is not possible to establish a link with the pre-OP values; secondly, it is a change in the function of the hand or hand. do not depict the wrist.
For statistics:
I recommend you mention the study design (case series) and perhaps also in Level of Evidenz (IV)
One problem of the Kaplan Meyer curve is to correctly classify the patients lost to follow-up or the deceased patients. There is a work by Murray DW at JBJS Br. from 1997, which defines a so-called Worst Case scenario, so that eventually two curves emerge, and the truth lies between these two curves. I recommend to add this scenario in Figure 3.
In addition, the primary endpoint is missing, obviously "revision for any reason".
In my opinion, the survival curve for the indication of the 95% confidence interval, which can also be calculated with small case numbers, is missing.
To the results: The statistics in tabular form. 3 I do not think it is necessary, even here, in my view, to specify the relative frequencies and the mean value when this complication occurred.
I think patient satisfaction is very important and well explained. I can only fail to understand the reference to pre-operative data, but it says elsewhere that this data does not exist.
I also consider the data on grip strength to be important, which I would elaborate on in a table, with mean value and standard deviation. There is also no problem that there are no data before the operation.
For the data on mobility before and after the operation, in my view, the range of movement is also sufficient, here the statistical evaluation is missing and the indication of the test with which this was done (T-test for connected samples).
Loss. 4 I cannot fully understand, I do not know what these data have to do with the function of the wrist prosthesis
To the scores: I do not think the SF 36 score is suitable, for the function of the wrist this score is not specific enough. That's why Tab can. 6 from my point of view too. I suggest that we focus on the VAS here. Here again: I would not give estimated values for the DASH score in a publication.
I think the X-rays are important and I would find an X-ray image helpful over the ten years as an image in the text.
For discussion: line 277: I would recommend to give here an exact percentage, with the confidence interval.
In addition, I find a table helpful to compare the long-term results of the work presented with the results of literature. In particular, it makes sense to list the revision rate or average duration of other prosthetics in order to make the comparison with other studies clearer to the reader and to support your conclusion.
I would leave out the discussion on the DASH score for the above reasons.
Concerning the limitations: The number of cases is not a limitation for the survival curve, it only increases the conference interval. The small number of cases is important for comparing mobility before and after surgery. In statistical terms, the study's power is too small to show statistically significant differences. I do not see any limitation when measuring the grip strength, you can easily compare it with the grip strength after the implantation of other wrist endoprosthetics. This would also be something for the results, as a table.
I would point out with regard to the restrictions that there is no reference to pre-operative data and that, after completing this Worst Case scenario and supplementing the CI, only a relatively rough estimate of the service life can be given.
Germany Version:
Sehr geehrte Autoren,
Ich erlaube mir folgende Anmerkungen zum Manuskript:
Ein generelles Problem des Artikels besteht meines Erachtens darin, dass die verschiedenen Teile, also Einleitung, Material und Methoden, Ergebnisse und Diskussion oftmals vermengt werden. Ich empfehle, die Arbeit dahingehend gründlich zu überarbeiten. Beispielsweise ist aus meiner Sicht auf Seite 1, Zeile 37 die Anmerkung zum erhöhten funktionellen Anspruch der Rheumapatienten Teil der Diskussion, als Rechtfertigung für die Indikation einer Endoprothese und gegen eine Arthrodese, und sollte nicht in der Einleitung genannt werden. Seite 2, Zeile 54 fortfolgende, gehört zum Material und Methoden, ebenso Zeile 71 fortfolgende.
Zu Material und Methoden: Ich halte Abbildung 2 und Tab. 1 nicht für erforderlich, meines Erachtens reicht hierzu Einsatz.
Bei Tab. 2 ist aus meiner Sicht die Angabe der relativen Häufigkeit völlig ausreichend.
Der Dash-Score wurde in der Arbeit abgeschätzt (Zeile 122)? Ich halte dieses Vorgehen nicht für sinnvoll, erstens kann kein Bezug zu den Werten Vor-OP hergestellt werden, zweitens ist damit die Veränderung der Funktion der Hand bzw. des Handgelenkes nicht abzubilden.
Zur Statistik:
Ich empfehle euch, das Studiendesign (Fallserie) zu nennen, und vielleicht auch in Level of Evidenz (IV)
Ein Problem der Kaplan-Meyer-Kurve besteht darin, die Patienten lost to Follow-up oder die verstorbenen Patienten richtig einzuordnen. Hierzu gibt es eine Arbeit von Murray DW im JBJS Br. aus 1997, die ein sogenanntes Worst Case Szenario definiert, sodass letztendlich 2 Kurven entstehen, und die Wahrheit zwischen diesen beiden Kurven liegt. Ich empfehle, dieses Szenario in der Abbildung 3 mit zu ergänzen.
Außerdem fehlt die Angabe des primären Endpunktes, offensichtlich „Revision aus jeglichen Grund“.
Es fehlt meines Erachtens bei der Überlebenskurven für die Angabe des 95 % Konfidenzintervalls, das sich auch bei kleinen Fallzahlen berechnen lässt.
Zu den Ergebnissen: Die Statistik in Tab. 3 halte ich nicht für erforderlich, auch hier reichen aus meiner Sicht die relativen Häufigkeiten, und die Angabe des Mittelwerts, wann diese Komplikation aufgetreten ist.
Die Patienten-Zufriedenheit finde ich sehr wichtig, und gut geschildert. Ich kann nur den Bezug zu den präoperativen Daten nicht nachvollziehen, an anderer Stelle steht, diese Daten gibt es nicht.
Die Daten zur Griff-Stärke halte ich auch für wichtig, die würde ich in einer Tabelle näher ausführen, mit Mittelwert und Standardabweichung. Hier ist es auch kein Problem, dass keine Daten vor der Operation vorhanden sind.
Bei den Daten zu Beweglichkeit vor und nach der Operation reicht aus meiner Sicht auch der Bewegungsumfang, hier fehlt die statistische Auswertung und die Angabe mit welchem Test das gemacht wurde (T-Test für verbundene Stichproben).
Tab. 4 kann ich nicht ganz verstehen, ich weiß nicht, was diese Daten mit der Funktion der Handgelenksprothese zu tun haben
Zu den Scores: Den SF 36 Score halte ich nicht für geeignet, für die Funktion des Handgelenkes ist dieser Score nicht spezifisch genug. Daher kann Tab. 6 aus meiner Sicht auch weggelassen werden. Ich schlage vor, hier den VAS in den Mittelpunkt zu stellen. An dieser Stelle noch mal: Geschätzte Werte für den DASH Score würde ich in einer Publikation nicht angeben.
Die Röntgenbefunde halte ich für wichtig und fände ein Röntgenbild im Verlauf über die 10 Jahre als Abbildung im Text hilfreich.
Zur Diskussion: Zeile 277: Ich würde empfehlen, hier eine genaue Prozent Angabe zu geben, mit dem Konfidenzintervall.
Zusätzlich finde ich eine Tabelle hilfreich, um die langfristigen Ergebnisse der vorgelegten Arbeit mit den Ergebnissen der Literatur zu vergleichen. Insbesondere macht es Sinn, die Revisionsrate oder die durchschnittliche Standzeit anderer Prothesen aufzuführen, um den Vergleich mit anderen Studien für den Leser übersichtlicher darzustellen und eure Schlussfolgerung zu untermauern.
Die Diskussion des DASH-Scores würde ich aus den genannten Gründen weglassen.
Zu den Limitationen: Die Fallzahl ist keine Limitation für die Überlebenskurve, sie vergrößert nur das Konferenz-Intervall. Für den Vergleich der Beweglichkeit vor und nach der Operation ist die geringe Fallzahl von Bedeutung. In statistischen Worten: die Power der Studie ist zu gering, um hierfür statistisch signifikante Unterschiede aufzuzeigen. Ich sehe bei der Messung der Griffstärke keine Limitation, Ihr könnt sie sehr wohl mit der Griffstärke nach der Implantation anderer Handgelenks-Endoprothesen vergleichen. Dies wäre auch etwas für die Ergebnisse, als Tabelle.
Ich würde bei den Limitationen aufführen, dass es keinen Bezug zur präoperativen Daten gibt, und dass nach Ergänzung dieses Worst Case Szenario und Ergänzung des KI nur eine relativ grobe Abschätzung der Standzeit angegeben werden kann.
Author Response
Dear Reviewer,
thank you very much for the detailed comments and the comprehensive suggestions for improvement. We have taken them on board and explained them point by point in an attached Word file in English and German. We hope that our revision has increased the chance of publication and that it meets with your approval.
Lieber Reviewer,
vielen Dank für die ausführliche Kritik und die umfassenden Verbesserungsvorschläge. Wir haben diese aufgenommen und sie Punkt für Punkt in einer anhängenden Word-Datei auf Englisch und Deutsch erläutert. Wir hoffen, dass mit unserer Überarbeitung die Chance auf eine Veröffentlichung gestiegen ist und diese Ihre Zustimmung findet.

Reviewer 2 Report
It's an very interesting topic, but I am afraid the data is not sufficient for publication.
There is no defined follow up, lots of data is missing. I wonder how many patients were eventually evaluated . In the abstract you claim 34 TWA, but in the results I can just find 16. That does not really fit.
The article is far too long - please shorten introduction and discussion. It's very hard to read and get out the information. You shoild work over it again to give a clear idea of what you wanna say.
In the present form it's not ready for publication.
Author Response
Dear reviewer,
we added our responding letter to your comments as a word document.

Reviewer 3 Report
The research is well designed and carried out.
Abstract: please remove paragraphs.
Introduction contains enough background informations regarding the techniques involved.
Materials and methods are clearly described.
Results: statistical analysis is carried out in a proper way, but the samples involved a small number of participants, I think that this could be considered as preliminary study: I appreciate that you state it as a drawback of your work in discussion.
Conclusions are ok.
Figures and tables are very good.
Author Response
Authors response to Reviewer 3 Comments: Thank you for your review and your kind comment on our article.
Point 1: Abstract: Please remove paragraphs.
Response 1: We have removed the paragraphs in the abstract.
Points 2-6:
Response 2-6: The other reviewers have made suggestions for improvement, which we have gladly taken up in order to further improve the manuscript. We hope that you can also agree with these revisions.
Round 2
Reviewer 2 Report
Manuscript improved.
Unfortunately the data is not sufficient. Actually itr's only 16 patients that could be investigated in the end. There are lots of patients with very short follow up.
The familiy doctor's records can not be the base for a scientific publication.
There is also a mix of indications- you should focus on one group- rheumatoid or non-rheumatoid as you know that those groups are clinically very different.
I am sorry but I still can't suggest this work for publication
Author Response
Response to Reviewer 2 Comments, round 2
Manuscript improved.
Authors response: Dear reviewer, thank you for your positive remark.
Point 1: Unfortunately, the data is not sufficient. Actually, it's only 16 patients that could be investigated in the end. There are lots of patients with very short follow up.
Response 1: I would like to thank you for once again addressing the central point of criticism of the study with the very small patient population. When planning and conducting the study, I did not sufficiently take into account the delay and the many losses due to death or relocation. Otherwise, I would have made the study period longer, which would have resulted in a larger number of operations. In retrospect, however, it is not possible to still include patients from 2005 to 2010, as I am no longer at the hospital and no longer have access to patient data and patient acquisition.
Point 2: The family doctor's records cannot be the basis for a scientific publication.
Response 2: I completely agree with you that scientific statements should not be made "on the call" of colleagues but should be collected personally based on previously defined criteria and scores. In the study presented, we explicitly point out this problem. We were only able to examine 16 patients personally, and our results are therefore statistically of limited reliability. On the other hand, our inquiries revealed that patients had died in the meantime or had their prosthesis explanted elsewhere. Consequently, the collection of follow-up data was not possible. However, these dropouts must be determined and taken into account in the context of the survival probability of the study. Otherwise, in our opinion, the results would be biased. The elimination of poor values/dropouts would be tantamount to falsification and is not a scientific act.
Point 3: There is also a mix of indications- you should focus on one group- rheumatoid or non-rheumatoid as you know that those groups are clinically very different.
Response 3: Thank you for your remark. I have checked the documentation again, regarding the serological status of the patients. I must apologize for the transmission error and have corrected the data accordingly (Line 73 to 77). So that you do not have to believe that I am manipulating patient data to my advantage here, I am attaching the data with solved names as an Exel file. Independent of our study, patients with negative serostatus may nevertheless belong to the group of patients with inflammatory disease. These patients sometimes show the same or comparable reactions to surgery and medication as rheumatics but differ significantly from osteoarthritic concerning their overall situation.
I am sorry but I still can't suggest this work for publication
Authors response: Dear reviewer, I hope you can agree to publication with the changes I made to the manuscript.
